# Spatial Distribution and Retention in Loblolly Pine Seedlings of Exogenous dsRNAs Applied through Roots

**DOI:** 10.3390/ijms23169167

**Published:** 2022-08-15

**Authors:** Zachary Bragg, Lynne K. Rieske

**Affiliations:** Department of Entomology, University of Kentucky, S-225 Agricultural Science Center North, Lexington, KY 40546-0091, USA

**Keywords:** RNA interference, root application, translocation, *Pinus taeda*, *Dendroctonus frontalis*, pest suppression

## Abstract

Exogenously applied double-stranded RNA (dsRNA) can induce potent host specific gene knockdown and mortality in insects. The deployment of RNA-interference (RNAi) technologies for pest suppression is gaining traction in both agriculture and horticulture, but its implementation in forest systems is lagging. While numerous forest pests have demonstrated susceptibility to RNAi mediated gene silencing, including the southern pine beetle (SPB), *Dendroctonus frontalis*, multiple barriers stand between laboratory screening and real-world deployment. One such barrier is dsRNA delivery. One possible delivery method is through host plants, but an understanding of exogenous dsRNA movement through plant tissues is essential. Therefore, we sought to understand the translocation and persistence of dsRNAs designed for SPB throughout woody plant tissues after hydroponic exposure. Loblolly pine, *Pinus taeda*, seedlings were exposed to dsRNAs as a root soak, followed by destructive sampling. Total RNA was extracted from different tissue types including root, stem, crown, needle, and meristem, after which gel electrophoresis confirmed the recovery of the exogenous dsRNAs, which were further verified using Sanger sequencing. Both techniques confirmed the presence of the exogenously applied target dsRNAs in each tissue type after 1, 3, 5, and 7 d of dsRNA exposure. These findings suggest that root drench applications of exogenous dsRNAs could provide a viable delivery route for RNAi technology designed to combat tree feeding pests.

## 1. Introduction

RNA interference (RNAi) is a pervasive eukaryotic genetic process that, when manipulated, can enhance our understanding of insect genetics [1,2,3], control insect-vectored viruses [4], and pathogenic fungi [5] as well as suppress plant-feeding insect pests [6,7]. The RNAi pathway can be triggered in insects and other organisms by introducing short hairpin RNAs (shRNA), small interfering RNAs (siRNA), precursory micro RNAs (miRNA), or double-stranded RNAs (dsRNA) [8]. The introduction of exogenous dsRNA is one of the most effective modes of RNAi initiation that can cause both gene silencing and mortality in insects [9,10,11]. Engineered dsRNAs are the key to RNAi’s specificity to target, as the mode of action relies on complementary sequences of genetic code to successfully silence translation, making this pathway considerably more specific than traditional broad spectrum insecticides [12]. As such, the control of insect pests using dsRNAs shows fewer off-target and environmental impacts when compared to traditional insecticides [13,14,15,16].

RNAi’s ability to induce host-specific gene silencing paired with reduced off target effects has led to its integration into pest management strategies for a variety of insect pests and pathogens with a multitude of deployment options [17,18]. dsRNAs have successfully induced gene silencing when applied using traditional topical methods including direct contact [19] and foliar sprays [20,21,22]. The RNAi pathway has successfully been induced via dsRNAs delivered systemically via trunk injection [23], whole-plant sprays [24], and root drenching [25]. In laboratory settings, the RNAi pathway has also been induced in insects feeding on transgenic plants expressing dsRNAs such as rice [26] and maize [27]. A transgenic maize plant incorporating dsRNA engineered to combat the corn rootworm pest complex (*Diabrotica* spp.) has been produced [28], and pending approval, this product could be one of the first commercially available commodities with intrinsic RNAi protection in the U.S. Moreover, field trials of sprayable dsRNA formulations have shown comparable results to traditional insecticides and this technology is being developed as the active ingredient in a consumer product for the control of Colorado potato beetle (*Leptinotarsa decemlineata*) [29]. These advances are bridging the gap between laboratory validation, field trials, and consumer availability and facilitating the integration of RNAi into pest management practices. While more research is needed to fully understand the complex interactions of dsRNAs in the environment, recent advances in the agricultural industry have moved RNAi from the laboratory to the field [30,31,32]. dsRNAs have been shown to induce the significant mortality of herbivorous crop pests [33] and regulate populations through reduced fecundity via phenotype alteration [27]. Numerous products have been combined with intrinsic or exogenous RNAi protection in herbaceous plants [26,34,35], but little work has been conducted looking at integrating RNAi into managing woody plant pests (but see Hunter, Glick, Paldi, and Bextine [25] for the treatment of citrus trees with dsRNAs).

In laboratory settings, several dsRNAs have shown efficacy in inducing the RNAi pathway in forest insects including the emerald ash borer (*Agrilus planipennis*) [36], Asian longhorned beetle (*Anoplophora glabripennis*) [37], and the southern and mountain pine beetles (*Dendroctonus frontalis* and *D. ponderosae*) [38,39]. Despite these advances, we lack a full understanding of the behavior and persistence of non-transgenic dsRNA, whether applied topically or systemically, in coniferous plant material (but see Pampolini et al. [40] and Bragg and Rieske [41] for work in deciduous plants). Therefore, we investigated how RNAi technology could be deployed against the southern pine beetle (SPB), a xylophagous, tree-killing beetle endemic to southeast and southcentral North America including Mexico. SPB larvae and adults carve winding tunnels through cambial tissue, disrupting the translocation of water and nutrients and effectively girdling their hosts [42]. While SPB can utilize most *Pinus* species, they prefer loblolly pine, *P. taeda* [43], which is the most economically and ecologically important pine in the southeastern U.S. Under normal conditions these beetles attack damaged or stressed trees, a behavior that plays a vital role in forest succession; however, when SPB reach epidemic or outbreak levels, they attack healthy trees and cause significant mortality over large, forested areas [44], with substantial economic and ecological impacts. Both southern and mountain pine beetle are undergoing unprecedented geographic range expansions in recent years [45,46], and on a global scale, *Dendroctonus* species are causing tree losses that significantly reduce localized carbon sequestration with cascading implications for climate change [47]. 

Forest pests such as SPB compromise forest productivity and sustainability and defy traditional management practices; additional management tools are clearly needed. In the laboratory, SPB demonstrates susceptibility to RNAi, suggesting that exogenous dsRNA application may be a useful suppression tool [39]. However, before RNAi can be deployed for tree protection, we need a greater understanding of the environmental fate of the dsRNAs including how exogenously applied dsRNAs will behave in woody plant tissue. Therefore, we sought to investigate the behavior of dsRNAs applied as a root soak to better understand the (i) translocation and (ii) persistence *in planta* using gel visualization. Loblolly pine was selected as our model conifer and the model target gene for our pest-specific dsRNA was the shibire gene in southern pine beetle. In insects, the shibire (*shi*) gene encodes the protein dynamin, which plays a role in clathrin-mediated endocytosis and vesicle fission [48], and when silenced using orally ingested dsRNA, has been shown to elicit significant mortality in SPB [39]. This is the first study to investigate *in planta* behavior and the recovery of exogenously applied dsRNAs in a conifer species. 

## 2. Results

### 2.1. Plant Material and RNA Recovery

To investigate transport and retention within conifer tissue, two different naked dsRNAs, dsSHI and dsGFP, were applied to loblolly pine seedlings as a hydroponic root soak. Following exposure, seedlings were sectioned into different tissue types (roots, stems, crowns, needles, and meristems), and subsequently processed for end-point PCR validation of exogenous dsRNAs (Figure 1).

Treated seedlings (*N* = 72) had an average root collar diameter (RCD) of 3.14 mm ± 0.09 (X ± SE) and an average height of 48.28 cm ± 0.63, and neither RCD (F_1,70_ = 1.38, *p* = 0.243) nor height (F_1,70_ = 0.121, *p* = 0.729) differed between treatments. Although seedlings originated from a single planting of one cohort, seedling size varied between replicates (RCD: F_2,69_ = 13.7, *p* = 9.4 ×10^−6^; height: F_2,69_ = 8.25, *p* = 6.1 ×10^−4^), with a significant interaction between the replicate and treatment for RCD (F_2,66_ = 3.53, *p* = 0.035) but not for height (F_2,66_ = 0.542, *p* = 0.584). These differences in seedling metrics can be attributed to natural variations in seedling growth over our experimental intervals, which ranged from March to June (rep 1: 16 March; rep 2: 19 May; rep 3: 1 June). We extracted total RNA from each of the five tissue types from each seedling, generating 360 RNA samples and recovering 170.8 ng/µL ± 5.25 RNA from each sample (~3417 ng). 

### 2.2. Recovery of Exogenous dsRNA

#### 2.2.1. Gel Recovery 

Exogenous dsRNA presence/absence was qualified by the presentation of a distinct product band matching the sequenced treatment products. Results from gel visualization fell into three broad categories: (1) those that showed bright, distinct product recovery in all tissue types (Figure 2a), indicating a strong “presence” of product; (2) those with mixed recovery, with some distinct bands and some with faint or no bands (Figure 2b); and (3) no product recovery (Figure 2c). While nearly every sample amplified a non-specific product (NSP) (Figure 2d), these were simply ignored as they were merely artifacts of the dsRNA primers used to generate the treatments. Instead, only the desired products were counted and analyzed, as confirmed by sequencing and size matching. Tissue samples from the control seedlings, exposed to water only (–dsRNA), showed successful amplification of the endogenous control sequence but no amplification of either dsRNA treatment.

dsRNA was successfully recovered from each of the five tissue types at least once in each of the four treatment intervals, but the recovery differed between dsRNA treatment, tissue type, and replicate. Proportional success of dsRNA recovery (no. samples with verifiable recovery via gel imagery divided by total no. samples) decreased as the exposure time increased, with day 1 having the highest recovery and day 7 having the lowest (Figure 3a); this pattern holds for both dsSHI (Figure 3b) and dsGFP (Figure 3c). Although not statistically significant, the recovery of dsRNAs differed slightly between the tissue types, with stem tissue yielding the most successful recovery, followed by the crown, then needle and meristem equally, and finally, the root tissue, having the lowest proportional recovery. A chi-square test of independence showed that there was no significant association between the recovery of exogenous dsRNA treatments and tissue type (χ^2^_4_, *N* = 360 = 2.1, *p* = 0.71). Individually, dsSHI showed the highest recovery in stem tissue, followed by the root, then crown, and finally, the needle and meristem equally whereas dsGFP had the highest recovery in needle tissue, followed by the meristem, then stem and crown equally, and finally, the root tissue.

Across all replicates, dsGFP was recovered from all five tissue types and present in all time points for 7 days. In contrast, dsSHI was recovered from each tissue type but only through the first 3 days. Collectively, seedlings in rep 1 had more instances of dsRNA recovery in both treatment groups when compared to the other two replicates (Appendix A). The differences between replicates were less drastic with dsSHI, where both reps 1 and 2 showed successful recovery of dsRNA at 1 and 3 days but none at 5 or 7 days and rep 3 showed a total lack of dsRNA recovery at all four time points. In contrast, in rep 1, dsGFP was recovered from seedlings at all four time points, but only recovered at one day in reps 2 and 3.

#### 2.2.2. Sequence Alignment

Alignment software was used to assign similarity scores based on the Myers and Miller global alignment algorithm to consensus reads in the treatment and recovered products (Table 1). Percent match was calculated by dividing the total matching base pairs by the total aligned base pairs excluding the non-aligned trailing sections. Treatment- and recovered-SHI had only a single base pair difference with an overall similarity of 99.71%. Treatment- and recovered-GFP products showed 100% similarity, with no-mismatches between the aligned segments.

#### 2.2.3. Logistic Regression Modeling

Across all tissue types, models that included all predictors (time, RCD, height, RNA recovery, replicate, and treatment) performed better than models that included only time as a predictor of dsRNA recovery. In the logistic regression model for each tissue type, the coefficients varied, thus the subsequent odds ratios and significance values also differed (Appendix A); however, models for all tissue types shared the same sets of significant predictors for dsRNA recovery. Time was the most significant predictor for the successful recovery of dsRNA, and for each tissue type, the odds ratios were <1.0, asserting a negative relationship between increasing time and successful recovery of exogenous dsRNA. Other factors that were significant for predicting the recovery of dsRNA were replicate, which was significant in all tissues except for the crown, and treatment, which was significant in the crown, needle, and meristem tissues. The remaining predictors, which included RCD, height, and RNA yield, returned non-significant *p*-values across all models. However, models that included all six predictors accounted for more variation, indicated by larger pseudo-R^2^ values, thus the final models included all characteristics (Figure 4).

## 3. Discussion

RNAi induced gene silencing is emerging as a next-generation pest management technique that could play a significant role in the forest management efforts of the future. The delivery of target pest dsRNAs is problematic and will likely be idiosyncratic, requiring practical pest- and site-specific prescriptions. One option under consideration in agricultural, horticultural, and forest systems is *in planta* delivery. Here, we confirm the presence of exogenous pest-specific dsRNAs applied hydroponically to experimental loblolly pine seedlings using gel visualization. 

Visualization using gel electrophoresis was our initial confirmation; this is contingent upon the recovery of total RNA and its subsequent reverse-transcription to complementary-DNA (cDNA). We recovered host plant RNA in the process of RNA extraction, but also denatured any dsRNA present in the plant tissue, generating single stranded RNA readily available for cDNA synthesis. Following the creation of unique cDNA, each sample was evaluated with primers corresponding to its treatment (dsSHI or dsGFP) to assess for the presence of exogenous dsRNA. Additionally, each sample was run with the endogenous control (*18s*) primer set to validate the success of the preceding RNA extraction, cDNA synthesis, and PCR sets. We used visualization to compare the sizes of the treatment products to the post-treatment recovered products (i.e., the PCR used to make the treatment dsRNA to the PCR products amplified from each sample using the dsRNA primers). In addition to yielding our target products, all three primer sets (dsSHI, dsGFP, and the endogenous *18s* control) yielded a smaller non-specific product (NSP) that, while distracting, had no effect on the successful visual identification of the desired product band for each gene. Although this is the first study to evaluate the translocation and persistence of exogenous dsRNA in coniferous woody plant tissues, previous work in ash (*Fraxinus* sp.) and oak (*Quercus* sp.) have shown the successful validation of dsRNAs using this visualization method [40,41].

Our sequencing results support the near identical matches of the treatment and recovered products and confirm that the products amplified from the total seedling RNA were the same as the exogenously applied dsRNA, allowing us to draw stronger conclusions from the gel images. Combined with the knowledge that tissue samples from seedlings treated with water only (–dsRNA) showed no amplification when paired with treatment primer sequences, we were able to confirm the presence of exogenous dsRNA using gel visualization. The presence of a product band affirmed the presence of dsRNA in that plant tissue at that time point, confirming the translocation and persistence of the dsRNA treatments. Additionally, sequencing also suggests that the NSPs in each product are likely primer dimers, as even in gel extracted and purified samples, these bands returned unreadable after sequencing. 

We found significant differences in dsRNA retention between replicates, and over the course of the 12-week study, our seedlings put on significant vertical and radial growth. Each replicate was conducted with fresh dsRNA synthesized from the same stock PCR products, and each replicate used the same greenhouse space and seedlings from the same cohort. The replicate effect we observed suggests that seasonal changes, possibly in localized plant growth rates, may play a role in the uptake and retention of dsRNAs in conifer plant tissues. Loblolly pine photosynthetic rates change seasonally [49,50], as do growth rates [50,51]. Many conifers have a single growth flush, but loblolly pine exhibits three distinct growth flushes in March, June, and July [50], accompanied by a height increase and new foliar growth. These timeframes roughly correspond to our experimental replicates (March, May, June), suggesting that the timeframe over which our experiment was conducted may have contributed to the observed differences between replicates. Although the seedling size differed between replicates, there were no within-replicate differences, and all experimental seedlings exceeded those designated as “plantable” and useable for research purposes in the southeast [52], supporting the potential application of our findings to RNAi as a tree protection strategy for loblolly pine. 

While differences in total RNA yield between tissue types were observed, total RNA yield was not a significant predictor for exogenous dsRNA recovery in any of our tissue models. Moreover, tissue with overall lower yields of total RNA recovery such as stem tissue showed greater proportional recovery of exogenous dsRNA, suggesting that RNA yield may not be a limiting factor in the recovery of dsRNA treatments using our methods. 

The effects of seedling tissue type on dsRNA recovery were not readily predictable. In pesticide applications, water-soluble pesticides present in different concentrations in different tissue types [53,54,55], and as such, the method and location of delivery influences the treatment concentrations in different tissues. In loblolly pine, specifically, the concentration of a water-soluble insecticide applied as a root drench remains most concentrated in the roots and is less concentrated in the distal tissues [56]. In contrast, with the recovery from all time points combined, our data suggest that the presence of dsRNA in the treated root tissue is no more likely than in other tissues including the stems, needles, and meristems, similar to the spatial and temporal distribution of exogeneous dsRNAs evaluated in white oak seedlings [41]. However, our methods here were semi-quantitative, and we were not able to determine the amount of dsRNA in each tissue type. In the future, quantitative analysis such as RT-qPCR could be used to quantify the spatial distribution of dsRNAs among tissue types.

We found dsRNAs in the seedling tissues not initially exposed to dsRNA, confirming within-plant movement; however, our methods provide no insights as to the mode of dsRNA transport through the tissues. Similarly, in deciduous oak, exogeneous dsRNAs disseminated throughout the seedling tissues, but the methodology did not provide insights into the mechanisms associated with their transport [41]. Herbaceous cucurbits (*Cucurbita* spp.) also showed systemic movement of dsRNAs from the surface treated leaves to the distal, untreated leaves, suggesting that after dsRNAs penetrate the leaf surface, internal vascular structures are responsible for within-plant movement [57]. In cotyledons of deciduous ash, Pampolini, Rodrigues, Leelesh, Kawashima, and Rieske [40] used fluorescently labeled dsRNA and confocal microscopy to verify the presence of exogenously applied dsRNAs and showed that the dsRNAs were present through the vascular tissues and within the inter-cellular spaces, but dsRNA was not found intracellularly, suggesting that dsRNAs mostly persist and move intercellularly in deciduous ash. Interestingly, in the current study, the mostly non-vascularized tissues, apical and lateral meristems, showed differences in dsRNA persistence between treatments. For seedlings treated with dsGFP, the meristem tissue had the second highest proportion of dsRNA recovery, while in the dsSHI treated seedlings, the meristem, along with crown tissue, had the lowest proportion of dsRNA recovery. The confocal imaging of deciduous plant tissue exposed to both dsRNA and hpRNA showed confinement to the xylem tissue, perhaps due to the size of the RNA molecules [23,40]. While larger RNA molecules may be confined to vascular tissue, the movement of sRNA molecules has been documented over long distances, mediated by vascular tissues, and over short distances, facilitated via plasmodesmata [58]. In our study, differences in the dsRNA size and subsequent transportability may have been a contributing factor in the greater abundance of dsGFP in non-vascularized seedling tissues such as the meristem. 

Interestingly, *in planta* behavior of our two treatment dsRNAs differed both spatially and temporally. In the dsGFP-treated seedlings, dsRNA products were recovered in more tissues and for a longer time than the seedlings exposed to dsSHI. Since dsSHI and dsGFP differ in size by nearly 130 bp (370 and 248 bp in length, respectively), size could play a crucial role in movement and persistence. Oak seedlings exposed to dsGFP as a root soak retained these dsRNAs in the distal leaves and stem in over 80% of samples at 7 d [41], showing a more robust retention of dsRNA than the conifers here. Differences in plant physiology, metabolism, and internal conditions are likely to be responsible for the differences in the recovery of similarly sized dsRNAs in coniferous versus deciduous seedlings. Plant RNases could potentially degrade the dsRNA, compromising integrity and preventing recovery with the semi-quantitative approaches we used. Plants contain a suite of dicer-like proteins (DCLs) [59] with specific enzymatic properties that cleave dsRNAs, each of which is confined to its own unique pathway, but with some redundancy. For instance, DCL4 in *Arabidopsis thaliana* has an affinity for substrates with long dsRNA sequences, while DCL3 prefers a shorter dsRNA substrate [59]. Exogenous dsRNAs, both naked or combined with clay nanosheets, can be processed by plant nucleases into siRNAs and have been shown to accumulate and provide viral resistance in non-treated plant tissues [21,60]. Despite plant processing, exogenous dsRNAs and their subsequent plant-generated siRNAs have been shown to persist up to three months post treatment when applied as a root drench to adult deciduous trees [25]. 

Initial work on conifer DCLs suggests that they facilitate important self-regulatory pathways in an effort to manage large genomes [61,62]. Conifers possess machinery that process RNA into 21-nt segments and appear to lack the protein structure that process dsRNA into larger 24-nt segments. DCL3 is associated with the generation of 24-nt segments, but is also noted for its affinity to shorter substrates. While “short” substrates are generally characterized as 50 bp or fewer, perhaps the lack of short-stranded preferring DCLs in conifers allows for shorter dsRNAs to persist longer. In contrast, longer dsRNAs may be more readily digested by conifer specific DCLs. Our dsRNAs were both longer than 50 bp, however, their size discrepancy of nearly 130 bp could explain differences in the degradation rates over time. dsGFP was recovered more consistently and for longer periods of time than the longer dsSHI. Future work looking at the persistence of dsRNAs of various sizes within plant tissues may inform the optimal sizing of pest-specific dsRNAs deployed within specific plant tissues and with specific tree species, but much remains to be learned. Additionally, the evaluations of dsRNA applications to seedlings in soil using traditional root drenching are essential to better understand how dsRNAs behave with a soil interface, and insect bioassays are needed to assess the dose required to induce gene silencing and subsequent mortality from dsRNA after exposure to active plant tissue. 

In the first two years of loblolly pine growth, the ratio of the relative growth rates between the needle and root tissue remains constant, but the short-term localized tissue growth rate rapidly changes seasonally [63]. Throughout the growing season, loblolly pine exhibit near monthly shifts in localized root and needle growth rates, drastically altering resource allocation and influencing cell division and growth. Bulk flow processes from source to sink are thought to be responsible for the majority of the long distance transport of dsRNA in plant tissue [64]. As such, fluctuations in localized growth patterns, and therefore subsequent altered flow of resources, may impact the transport of dsRNAs within tissues. While water and other resources flow throughout seedling tissues, allocation to areas of the highest growth may be the path of least resistance for exogenous dsRNAs. Therefore, in times of localized root growth that occur intermittently throughout the season, root applied dsRNAs may be held within the root tissues rather than distributed throughout the seedlings. Moreover, DCLs in conifers are believed to play a role in small RNA-mediated regulation of heterochromatin [61] and other vital organization processes associated with large genomes. As such, these DCLs may play an important role in gene expression, regulation, and other processes that could be impacted during cell division and expansion. Therefore, while unexplored here, tissues associated with accelerated rates of cell division and expansion may possess larger quantities of DCLs and therefore may prove more recalcitrant to the long-term persistence of exogenous dsRNAs developed for pest suppression. 

## 4. Materials and Methods

### 4.1. Plant Material

Following 30 d of damp stratification at 4 °C, loblolly pine seeds (7-56×OP and 20-1010, USDA Forest Service) were individually seeded into a 2:1 ratio of pine bark fines soil conditioner (Barky Beaver, Moss, TN, USA) and Promix general purpose growing medium BX (Premier Tech Horticulture, Rivière-du-Loup, QC, Canada) in 6.35 × 25.40 cm D40H deepot tree growing cells (Stuewe & Sons Inc., Tangent, OR, USA) during February 2019. Seedlings were maintained in the greenhouse (~18–22 °C, 15:9 L:D, 40–60% RH) and watered daily through seedling establishment, and then as needed. 

### 4.2. Target Gene Selection

Candidate gene selection was based on published reports, and a 370 base-pair double-stranded *shi* RNA (dsSHI) engineered specifically for SPB [39] was chosen as a treatment dsRNA. A second dsRNA treatment of green fluorescent protein (dsGFP, 248 bp), which is often used as a negative control for insect gene expression studies [65,66,67], was also selected. The differing lengths of our treatment dsRNAs can help inform us on how dsRNA size may impact translocation and persistence within conifer tissues. 

### 4.3. dsRNA Synthesis

Adult SPB were obtained from loblolly pine bark samples collected in summer 2019 from areas with outbreak populations in the southeast USA (Louisiana and Georgia). Total RNA was harvested from adult SPB within 24 h of beetle emergence using TRI Reagent RT (Molecular Research Center Inc., Cincinnati, OH, USA). Following RNA precipitation, integrity was verified by measuring absorbance at 260/230 nm and 260/280 nm. cDNA was synthesized using SuperScript¨ III Reverse Transcriptase (Invitrogen, Carlsbad, CA, USA) according to the manufacturer’s instructions and using protocols adapted from Kyre, Rodrigues, and Rieske [39]. To amplify the dsRNA template transcripts, polymerase chain reactions were run using primers specific to the gene of interest (Appendix A) using the following parameters: 4 min at 94 °C, followed by 35 cycles of 30 s at 94 °C, 30 s at 60 °C, and 45 s at 72 °C, and a final incubation step at 72 °C for 10 min. PCR product templates were purified using a Qiagen Purification Kit (Qiagen, Germantown, MD, USA), and the MEGAscript RNAi Kit (Thermo Scientific, Waltham, MA, USA) was used to synthesize dsRNA. Following incubation at 37 °C for 16 h, dsRNA was recovered by adding 0.1 × volume sodium acetate and 2.4 × volume of 100% ethanol and incubating at −20 °C for 2 h. After incubation, the mix was centrifuged (20,000× *g*) at −4 °C for 30 min, washed with 750 μL of 75% EtOH, and centrifuged again (18,800× *g*) at −4 °C for 15 min. After the ethanol wash, samples were dried at 37 °C for 25 min and re-suspended in 20 μL of nuclease free H_2_O. The quality of dsRNA was checked using electrophoresis and quantified with a spectrophotometer (NanoDrop Technologies, Wilmington, DE, USA).

### 4.4. dsRNA Exposure

Seedlings were removed from the potting medium and the roots were rinsed with tap water, followed by rinsing for 30 s with dd H_2_O before transferring to autoclaved 25 mL clear glass assay cylinders (1.5 cm × 12.5 cm). Seedlings were randomly assigned to treatments consisting of either 250 ng dsSHI or 250 ng dsGFP (*n* = 12); control seedlings received no dsRNAs (*n* = 3). dsRNA treatments were pipetted onto the wall of each assay cylinder, after which nuclease free H_2_O was added to a total volume equal to 25 mL with root volume displacement (Appendix A). Assay cylinders were topped with aluminum foil sheets to minimize water loss due to evaporation. dsRNA solutions were synthesized simultaneously, however, the concentration and subsequent treatment volume varied between the treatments and replication. Within each dsRNA treatment (dsSHI and dsGFP), seedlings were randomly assigned to experimental exposure times of 1, 3, 5, and 7 d (*n* = 3), while control seedlings (–dsRNA) were assigned to a 9 d exposure interval (*n* = 3). Seedlings (*N* = 27 per replicate) were maintained in the greenhouse; those seedlings designated for a given exposure interval were maintained together to optimize space and rotated daily to minimize any abiotic irregularities. Nuclease free H_2_O was also added daily to maintain a total volume of 25 mL (Figure 1).

### 4.5. Plant Processing

#### 4.5.1. Tissue Sectioning

Following dsRNA exposure, seedlings were rinsed to remove any excess treatment material and the total seedling length from the tip of the tap root to the apical meristem (hereafter “height” (cm)) and root collar diameter (RCD) (mm) were measured. Seedlings were then sectioned into: (1) roots, tip of the tap root to the root collar; (2) stem, root collar to the first needle; (3) crown, stem tissue from the first needle to the base of the shoot apical meristem; (4) needles; and (5) meristem, shoot apical meristem, and lateral meristem(s) if present. Sectioning tools were sterilized between tissue types. Seedling size was evaluated using a two-way ANOVA with Type II SS; Tukey’s HSD was used to determine the differences in seedling height and RCD for each time point and between each replicate. Seedlings designated as controls were not included in the statistical analyses.

#### 4.5.2. Tissue Homogenization & RNA Recovery 

Tissues were ground to a fine powder using liquid nitrogen and a mortar and pestle, and ~200 mg of each tissue type was transferred to a 1.5 mL microcentrifuge tube and stored on liquid nitrogen until RNA extraction. Following tissue homogenization, total RNA was extracted using a protocol modified from Chang et al. [68] (Appendix B). RNA pellets were resuspended in 20 μL RNase free H_2_O.

#### 4.5.3. RNA Quantification

RNA pellets were resuspended for 30 min at 21 °C with 5 s of light vortexing every 10 min. Once completely resuspended, the RNA concentration and purity were analyzed via absorbance measurements of 260/230 nm and 260/280 nm (NanoDrop Technologies, Wilmington, DE, USA). The concentration of each sample was recorded in ng/µL and the absorbance measurements were recorded as ratios. RNA samples were stored at –20 °C until further analysis. RNA recovery was evaluated using a two-way ANOVA with type II SS; Tukey’s HSD was used to determine differences among tissue type, time interval, and replicate. Initial analyses included all samples, however, the RNA samples from seedlings designated as the controls were not included in the final statistical analyses as the control seedlings were used only to verify the absence of treatment dsRNAs in the untreated tissue.

### 4.6. Analysis of dsRNA Presence

#### 4.6.1. cDNA Synthesis and PCR Amplification

Following RNA recovery, cDNA was synthesized using the SuperScript III Reverse Transcriptase protocol with a template of 500 ng of RNA, unless the initial concentration of RNA was not suitable (28 of 405 samples), in which case the maximum template size was used (Appendix A). Specific transcripts of interest were amplified from 1 µL of cDNA using a PCR protocol with modified incubation parameters consisting of an extended cycle count: 4 min at 94 °C, followed by 40 cycles of 30 s at 94 °C, 30 s at 60 °C, and 45 s at 72 °C, and a final incubation step at 72 °C for 10 min. Each cDNA sample corresponded to a unique combination of tissue type (root, stem, crown, needle, meristem), dsRNA treatment (dsSHI, dsGFP, or—dsRNA), and time point (1, 3, 5, 7, or 9 d). Each cDNA sample was used as the template in two PCRs, the first set of primers was specific to the treatment dsRNA that the seedling received, and the second set of primers corresponded to an endogenous control. The gene encoding 18S ribosomal RNA is both stably and highly expressed throughout the loblolly tissue types and developmental stages and thus selected as an endogenous control to verify the success of RNA extraction, cDNA synthesis, and PCR amplification [69]. For example, dsSHI primers were used on tissue samples from seedlings exposed to dsSHI, followed by primers corresponding to *18s* (Appendix A). cDNA samples from the control seedlings, treated only with water, had three PCRs, each with the dsSHI, dsGFP, and *18s* primer pairs.

#### 4.6.2. Gel Electrophoresis and Visualization

After amplification, the PCR products were combined with fluorescent loading dye and placed into a 1% agarose gel. Products were visualized using an UV Transilluminator and photographed; the presence of an amplicon matching the size of the treatment dsRNA equated to the successful recovery of exogenous dsRNA in each tissue type. Differences in recovery by tissue type were assessed with Pearson’s chi-squared test.

The recovery of dsRNA in each sample, as verified by gel images, was treated as a binary dependent variable, where successful recovery was equal to 1 and unsuccessful recovery 0. A logistic regression model was selected to estimate the factors that influenced the successful recovery of dsRNA in the treated tissues. Separate models were created for each tissue type wherein each recovery of exogenous dsRNA served as the response variable, and the time, RCD, height, recovered RNA concentration, replicate, and treatment all served as predictors. In each model time, RCD, height, and RNA concentration were considered as continuous variables, and the replicate and treatment were treated as categorical.

#### 4.6.3. Product Sequencing

A subset of genetic material was sequenced to confirm the presence of exogenously applied dsRNA within the plant material (Appendix A). A representative group of PCR samples generated by the previously described methods were run on a gel and subsequent products were extracted using QIAquick Gel Extraction Kit (Qiagen, Germantown, MD, USA). Once extracted and purified, an additional PCR was run to obtain material for genetic sequencing. Six PCR samples (Table 2) were Sanger sequenced by the University of Kentucky Health Center Genomics Core Laboratory (Lexington, KY, USA) using a 16-capillary 3130XL sequencer. Sequences from the treatment and recovered material were compared to verify the presence of exogenously applied dsRNA. Consensus reads were generated using the sangeranalyseR package for R [70]. Pairwise nucleotide sequence alignment for taxonomy tool (EzBioCloud, ChunLab Inc., Seoul, Korea, 2017) was used to align consensus reads from the treatment and recovered pairs, and the similarity scores based on the Myers and Miller global alignment algorithm were used to verify the presence of the treatment genetic material in post treatment seedling tissue.

## 5. Conclusions

RNAi is rapidly emerging as a next generation biopesticide that has already been deployed for field crop and horticultural pests; however, its deployment against tree pests is lagging, due in part to difficulties associated with delivery. Here, we document the successful uptake and retention of exogenous SPB-specific dsRNAs applied hydroponically in loblolly pine seedling tissues; this is the first study to evaluate the spatial and temporal distribution of exogeneous dsRNAs engineered for insect suppression in conifers. Previous work has evaluated dsRNAs in woody plants, but those investigations evaluated the different means of delivery [23], greater amounts of dsRNA [25], younger plant material on smaller scales [40], or focused solely on a deciduous species [41].

Our current study elucidates the spatial and temporal behavior of dsRNAs in pine seedlings and suggests that for pest suppression, it will be important to consider the pine growth rates and seasonal changes in plant metabolism for dsRNA uptake and within-plant distribution. Additionally, an understanding of the effects of the pest-specific dsRNA size on the degradation rates could be important for within-plant persistence. Undoubtedly, additional factors will require careful study and consideration, but this work helps lay the foundation for future research exploring the deployment of dsRNA engineered for insect pest suppression in conifers. Our methodology proves that dsRNA can be introduced to tree seedlings hydroponically, and recovered and amplified, even when the initial dose is incredibly small. While larger doses may be necessary to silence genes and cause insect mortality, this work shows that even small amounts of dsRNA rapidly disseminate throughout the conifer tissue and may remain in different tissues for extended periods of time, contributing to tree protection. This lays the foundation for understanding how dsRNAs might be deployed for single tree protection using RNAi, contributing to a strategy for next generation pest suppression.

## Figures and Tables

**Figure 1 ijms-23-09167-f001:**
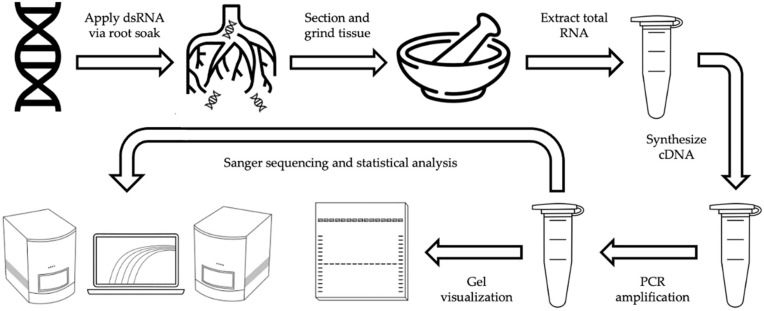
The experimental workflow from the hydroponic application of dsRNA treatments to seedling roots through sequencing validation and statistical analyses.

**Figure 2 ijms-23-09167-f002:**
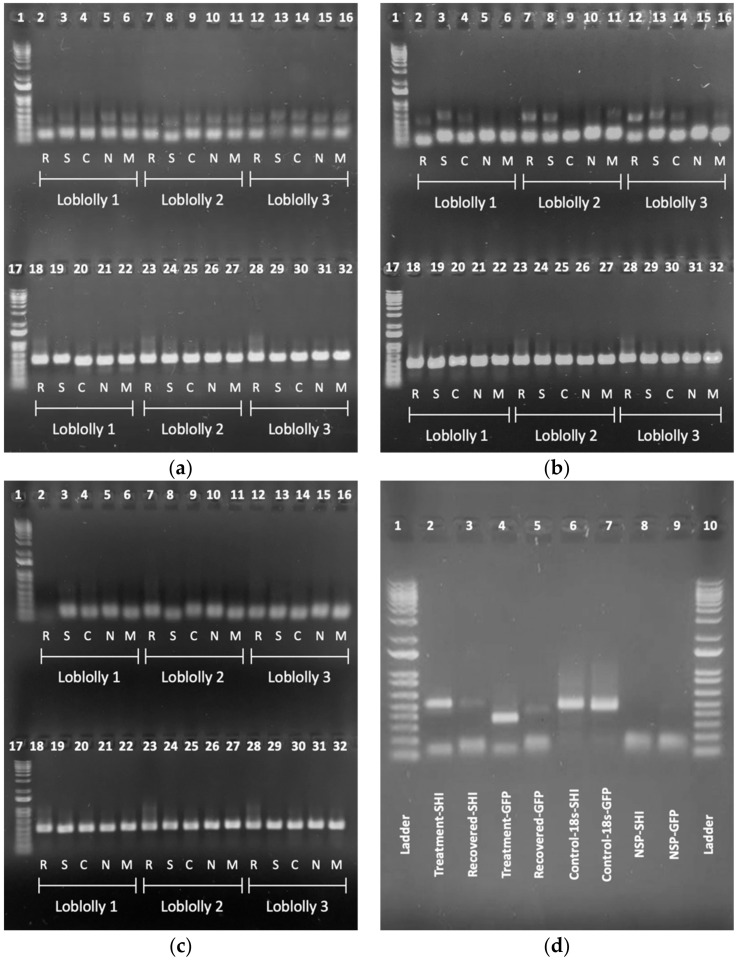
Gel images of PCR amplification from the first replicate of loblolly pine seedlings exposed to: (**a**) 250 ng dsGFP, (**b**) 250 ng dsSHI on day 1, and (**c**) water (–dsRNA) on day 9. Wells 1 and 17 contain a 1 kb DNA ladder; wells 2–16 contain samples amplified using either dsGFP (**a**) or dsSHI (**b**,**c**) primers, where the upper amplicon corresponds to the desired product and the lower amplicon to non-specific products; wells 18–32 contain samples amplified using *18s* primers, each with a single amplicon corresponding with the 18s primer set. (**d**) Gel image of treatment and recovered dsSHI (wells 2 and 3), treatment and recovered dsGFP (wells 4 and 5), endogenous controls (wells 6 and 7), and non-specific product (NSP) formation (dsSHI well 8, dsGFP well 9). Wells 1 and 10 contain 1 kb DNA ladder. R = root, S = shoot, C = crown, N = needle, M = meristem.

**Figure 3 ijms-23-09167-f003:**
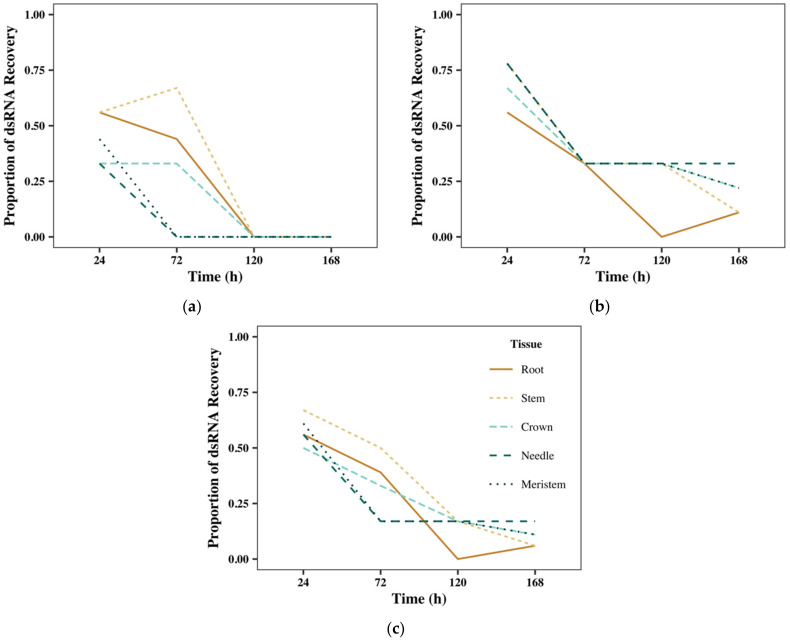
The proportional success of dsRNA recovery in loblolly pine tissues showing (**a**) combined dsSHI and dsGFP recovery, and the recovery of (**b**) only dsSHI, and (**c**) only dsGFP across replicates.

**Figure 4 ijms-23-09167-f004:**
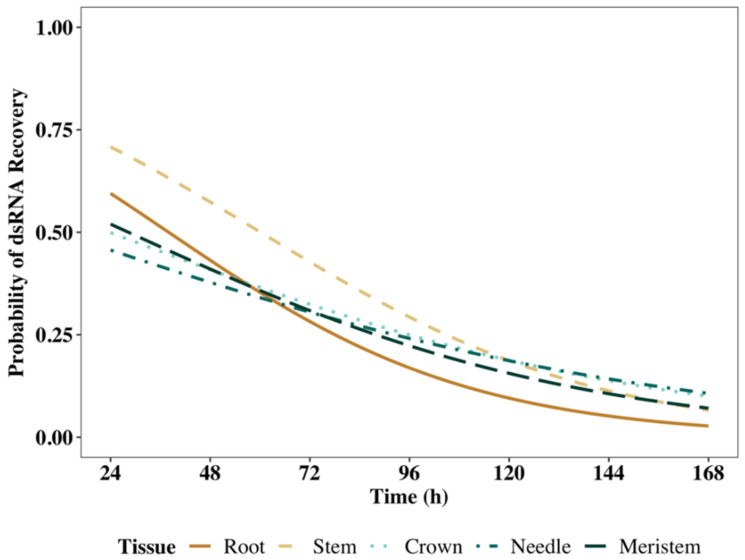
The projected probabilities of dsRNA recovery based on the logistic regression models for each tissue type through time. Models used to generate these projections are summarized in Appendix A.

**Table 1 ijms-23-09167-t001:** The sequence alignment data for the treatment and recovered product pairs.

Sequence 1	Length	Sequence 2	Length	Matches	Errors	Total	Match
Treatment-SHI	390	Recovered-SHI	380	346	1	347	99.71%
Treatment-GFP	267	Recovered-GFP	248	209	0	209	100.0%
Control-*18s*-SHI	251	Control-*18s*-GFP	258	214	1	215	99.53%

**Table 2 ijms-23-09167-t002:** The unique genetic products from loblolly pine seedlings treated with dsRNAs that were Sanger sequenced. Amplicon names are categorical references to different types of genetic materials.

Amplicon Name	Description
Treatment-SHI	PCR product used as the template to make dsSHI
Recovered-SHI	PCR product amplified using dsSHI primers on root tissue treated with dsSHI
Control-*18s*-SHI	PCR product amplified using *18s* primers on root tissue treated with dsSHI
Treatment-GFP	PCR product used as template to make dsGFP
Recovered-GFP	PCR product amplified using dsGFP primers on root tissue treated with dsGFP
Control-*18s*-GFP	PCR product amplified using *18s* primers on root tissue treated with dsGFP

## Data Availability

Data are available upon reasonable request to the corresponding author.

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
