# Peer review of "Spatial Distribution and Retention in Loblolly Pine Seedlings of Exogenous dsRNAs Applied through Roots"

_ijms, 2022, doi:10.3390/ijms23169167_

Round 1
Reviewer 1 Report
Comments:
- Rendering of Figures 1,3,4 is poor and severely detracts from the paper. Consider requesting a vector graphic if possible, if not use a different image format or reduce the size of the image.
Author Response
We have generated high resolution figures and uploading them in the revision. Thank you for the review.
Reviewer 2 Report
The authors addressed all my concerns very well. I have no more comments on the revised manuscript and recommend its publication.
Author Response
Thank you for your review.
This manuscript is a resubmission of an earlier submission. The following is a list of the peer review reports and author responses from that submission.
Round 1
Reviewer 1 Report
Major points:
- Include some reference to potential siRNA longevity in discussion.
- Plants have RNA dependent DNA & RNA polymerases and can possibly retrotranspose dsRNA into their genome or amplify expression of dsRNA, i.e. parts of the dsRNA can be incorporated but it could be a sequence specific mechanism. For this reason and others, don't combine treatment groups for analysis if you can avoid it, it looks like you are scraping the barrel and it's really clear that there's variation in what comes back in different tissues. I think it's interesting how rapidly dsRNA is 'cleared' and how it varies between tissues is an interesting point of discussion. Combining them to look stronger actually makes the paper weaker because you are assuming similarities in sequences which isn't true. Also, it would be interesting to see how long dsRNA secondary products, degraded products, and siRNAs stick around for, could have done Northern blotting to see that or siRNA-seq.
- Results section starts but doesn't really give any information about what the groups are in the experiment and pushes the study design figure to the very bottom of the manuscript. Please include a results section going over the study design in the results to include the figure and also to point out and clarify the groups, dsGFP, dsShi, untreated so the audience is aware. It's all very vague otherwise at the start of the results
- Please flip order of results, it's weird to see the final results first then work backwards to gel images, when gel images were done first. Why not order as such? Do (1) study design, (2) gel images, (3) sequencing, and (4) melt curves, then (3 or 5) regression models which can still come at the end.
- For table 1, why not make a consensus sequence combining F and R, use the consensus sequence to determine differences? This would be the best way to actually compare reads rather than specifically looking at F and R independently. You can always look at the chromatogram to pick the dominant peak if it exists.
- I don't really know why melt curve and sanger sequencing are both needed. Is it a proof of principle to identify dsRNA? If so make sure this is clear in the results section.
- Figure rendering is all messed up in places, see figure 2C, please organise proofs properly!
- Include all kits and companies used with details about manufacturers protocols in the methods (see lack of cDNA method as example).
Line edits below:
30: If you can please stick to or include Coleoptera examples in first sentence. There's a massive amount of data regarding RNAi in insects and we know now that differences between insect orders not only exist but are in cases not relatable to one another (e.g. SID-1 and SID-2 channels, systemic RNAi differences, uptake from the midgut, etc). This is just me being picky so rebut if needed! I generally like the flow of the intro.
36: There are definitely better and more recent papers to talk about in relation to dsRNA activation of RNAi in insects than the Fire paper. Please update or expand this reference.
69: what does systematically mean here? Injection and/or root soak? I guess gene editing can be system wide expression but can also be tissue specific. Please expand and clarify exposure routes here.
86: MPB - just say mountain pine beetle, there's no other use of MPB in the manuscript except when you spell it out on lines 66 and 80.
95: Can you please add 1 more sentence or expand to say that Shi was one of the most potent insecticidal RNAi targets from your previous work and that (I'm assuming) is why you chose it for this study.
99: root collar diameter is not defined as RCD until part 4 (materials and methods are below results) so please define on line 99 as it's the first appearance in the manuscript. Redefine in the methods for those who skip right down there.
100 + various places throughout: can you substitute p > 0.05 to the actual p value. It might not be statistically relevant but it's nice to see p = 1 rather than p = 0.051. I was never a fan on 1 in 20 being the cure all for every statistical methodological validation (no idea if plant growth under stress is normally distributed or not though, but lets not go there). Are these all in Table 2? Ignore my comment if so.
108-127: This whole bit is kind of dry, I never see 260/280 ratios in a results section of papers. I would never expect anyone to require the same RNA yield between samples, especially given you are introducing RNA to the plant, and that plants have individual differences. I would especially expect a ton of differences when you are doing a manual chemical extraction and not a filter column based kit. I would push a lot of this information to the materials and methods section because it's not really a result but just QC.
As for the 260/230 readings, you are always going to get salts and carryover so it's not as important as 260/280, but people don't even like nanodrop anymore, so discussing these measures in detail will just highlight the fact you should be doing Qubit, RIN (Bioanalyzer), and/or integrity gels. I would also say, unless you did 3x nanodrop readings per sample, you won't get the actual average (because salt or other carryover will mess with the reading and some vary quite a bit). Without an RNA integrity gel, the purity and concentration are relatively meaningless too. Also, if dsRNA is carried over, the concentration between samples is also impossible to gauge which is a MAJOR flaw since samples can't be standardized for cross comparison of RNA uptake (only when it comes to quantifying relative amounts). What you would need to do here is run a gel, measure the brightness of your rRNA bands and base concentration on the band intensity in relation to the nanodrop reading. A pure sample, with no smearing (other than mRNA), can be normalized to rRNA band intensity. All of these points are moot since you already did the study and it's not that important really, but when you write extensively about the RNA yield and quality, you just are likely to draw criticism.
146: There's no a,b,c lettering in the figure! Please clearly label figure parts and make them align to the figure legend. Poor presentation here. Other letters also not specified here! Put a R = root, C = crown, etc. in the fig legend.
147: why is water control on day 9? Looks pretty suspicious that you've got day 1 for exposure and day 9 for water. If RNA/cDNA is in the freezer still I would re-run day 1 PCR for this gel. Ultimately it's a control, but why the different time point here, please explain.
182: no figure labels again!
425: This needs to be figure 1! It's the whole overview of the paper and it's sitting way at the bottom, please make figure 1 and include a sentence or sub-heading in the results to say 2.1 Study design or something like that.
459: What cDNA kit was used, was it from ssRNA? If so this could be interesting point since you might not actually recover full yield of dsRNA here, unless properly melted, which I'm assuming is done prior to primer binding (70+ C) but depends on your kit that you used. Please specify.
462: using 'a' PCR protocol.
Reviewer 2 Report
RNAi through host plants has been recognized as a powerful tool for developing novel pest control technology, and exogenous dsRNA provides an alternative method for specific gene silencing and pest control. However, many studies have just focused on agricultural and horticultural pests, with a lack of application in forest insects, especially the movement of exogenous dsRNA through woody plant tissues, which is still unclear. The current study discovered that dsRNA molecules can be transported across different pine tissues by the root soak method. The paper's findings are quite valuable for understanding exogenous dsRNA movement throughout woody plants. However, the following suggestions should be carefully considered before acceptance.
Major comments
- Exogenous dsRNA delivered by plant soaking is a convenient method for gene silencing studies in insects, and the current paper also provided powerful evidence of the rapid movement of exogenous dsRNA by root soak. However, several publications have indicated that dsRNA is very unstable in the soil and will degrade rapidly and lose its biological activity within about 2 days, regardless of initial dsRNA concentration and soil differences, which suggests that it may be impractical to control insect pests through the root drench method of exogenous dsRNA in a real plant living environment surrounded by soil. Therefore, it is difficult to achieve the conclusion that fighting tree pests mediated by root drench application of exogenous dsRNA according to the current paper's main results. I suggest the authors rearrange the paper to make their findings' conclusion more reasonable and practical.
- The mode of exogenous dsRNA movement in pine seedlings is independent of the source of dsRNA, whether that dsRNA derives from insects or other organisms. Thus, I think an alternative title may be proper for the paper's main discovery: spatial distribution and retention in loblolly pine seedlings of exogenous dsRNA by root soak. It is recommended that the authors rewrite the relevant texts to make them more consistent with the real findings of the present study.
Minor comments
- All figures' quality should be improved and examined, and more detailed information in figure legends should be provided for the readers to easily understand. For example, in fig.1, no labels of a,b,c, and d letters in the gel image; and a lack of description of abbreviated letters, such as R/S/C/N/M above the "Loblolly" word in the gel image; no statistical information in both fig. 2 & 3.
- The authors found that shorter dsGFP (248 bp) persisted better than longer dsSHI (379bp) in more plant tissues and longer duration owing to the possible size effects. Because of the different sizes of dsRNA, more dsGFP molecules should be transferred to the plants than dsSHI with the same dose, so it could be due to the number of dsRNA rather than the size of dsRNA. Further experiments should be provided to verify the authors' hypothesis, such as using a similar number of dsRNA (based on mol) with different sizes to target the same genes.
Reviewer 3 Report
This manuscript seeks to assess the potential for delivery of dsRNA into a conifer, loblolly pine, using a root soak approach. The strategy used two different dsRNAs (an insect sequence and GFP) and involved detecting the presence of the two molecules in different seedling tissues, after different exposure times. The method of detection was end-point PCR. On the positive side, the manuscript is well written and clearly presents the data that are in hand. However, I have significant doubts about some of the approaches used and the results presented are accordingly speculative.
Firstly, the three methods of assessing the success of end-point PCR amplification are redundant. Only gel analysis is needed to determine if template was present. Melt curve analysis and sequencing only contribute in determining if any product detected was genuine, and of the two, melt curve analysis provides no additional useful information beyond sequencing. The sequencing results themselves are troubling. The number of mismatches with the dsRNA templates is very high; polymerase errors should not be producing up to 11% mismatches (Table 1).
Tangentially, the design of the study is pushing the limits of sensitivity of the technique of end-point PCR for detection of dsRNA in the plant tissues. Since 40 cycles of amplification were needed to see any specific products, primer-dimer formation is a significant competitor in the cycling process. Were the amplicon efficiencies measured? It appears from Fig 1 that the shi gene product amplicon is least efficient given the very high quantity of primer-dimer products in these lanes. This could suggest a reason why the shi gene product was detected far less often than the gfp product was. Ideally, a more sensitive technique like digital PCR would be preferable for detection of such low levels of product, or at least qPCR. Selecting more efficient amplicons for amplification from within the templates could also be helpful if end-point PCR is unavoidable.
The detection methodology aside, there is also the question of template generation. In the Discussion (lines 224-226) it is stated that “We recovered host plant RNA in the process of RNA extraction, but also denatured any dsRNA present in the plant tissue, generating single stranded RNA readily available for cDNA synthesis”. In fact, dsRNA molecules of the sizes used here will reanneal upon resuspension. Most first strand cDNA synthesis kits use a 65 degree denaturation step to prepare RNA for primer annealing because most RNA is single-stranded. Although no kit details were given in section 4.6.1, it is noted in section 4.3 that a Superscript III kit was used during template preparation. Assuming that the same kit was used for cDNA synthesis for the test samples, this would suggest 65 degrees was used here as well? Without a high temperature incubation, the dsRNA targets are not likely to prime consistently for first strand synthesis. Double-stranded RNA molecules of the sizes used here, would not be denatured at 65 degrees, and therefore cDNA synthesis would not be consistently representative of template quantities. Any additional details on the cDNA synthesis protocol used that could address this issue should be provided. It is also possible that greater sensitivity of the assays overall could be achieved by ensuring optimal cDNA synthesis from dsRNA recovered from the plant tissues.
Regarding the conditions used for the assays, on line 524 the text states “Our current study elucidates the spatial and temporal behavior of dsRNAs in pine seedlings…”. The temporal behavior can’t properly be evaluated without knowing the exposure time of the tissues. Was the integrity of the dsRNA in the drench solution assessed over the time period of the assay? I can’t find any indication this was evaluated. If it degraded over the 7 days of exposure, then the actual exposure time would be variable according to the decay of each template in each sample drench. An indication of the half life of the templates in the drench solutions would be essential to knowing the exposure time and therefore the temporal (and spatial) behavior of the dsRNAs.
Given the various uncertainties regarding the consistency of the detection results, regression modelling seems speculative. As an aside, I don’t see why Table 2 only includes data for dsGFP?
Specific points:
Title - the title is misleading, it implies a significant role of RNAi in the work performed in the study.
Line 95 - some information about the shi gene would be useful.
Line 126 – notes low 260/230 nm absorbance values were observed for some RNA samples. Line 276 speculates that the cause of the low absorbance values is compounds in the plant tissue. However, there are many other potential contaminants that could carry over from the extraction materials, and these could compromise cDNA synthesis efficiency. Was efficiency tested?
Fig 1 – the panels are not labeled.
Fig 2 – the panels are not labeled.
panel B - the meristem line is not visible in this panel.
Panel C – the chart is stretched
Line 221 – as mentioned previously, the authors have not used 3 approaches; they have used 1 approach, end-point PCR, and then used 3 methods to assess the results. This should be reworded.
Line 223 – this is actually reverse transcription.
Lines 320-323 – the differences in the presence of dsRNA for GFP and the shi gene in different tissues and over time could be due to the different product lengths as the authors discuss. However, it could also be the result of different stabilities of the dsRNAs in the drench solution, as mentioned above. Was dsRNA integrity in the drench solution tested?
Line 392 – states “Following RNA precipitation, integrity was verified by measuring absorbance at 260/230 nm and 260/280 nm.” These ratios measure purity, not integrity, of the RNA.
Line 396 - please specify where the primer sequences are given.
Line 445 – the protocol in Appendix A specifies a different volume of water for resuspension.
Lines 461-464 - How much cDNA was used as template in the end-point PCR?
Round 2
Reviewer 3 Report
On the positive side, the revised version of this manuscript has been significantly condensed to remove extraneous material and is much better focused. However, based on the author’s responses, the basic problem remains that the templates used for the end-point PCR amplifications will not be representative of dsRNA quantities in the extracted samples since the conversion of dsRNA target to cDNA is not quantitative without full denaturation of the strands before primer annealing. Compounding this variability in the template, the PCR amplicons used for the end-point PCR (which were taken from another paper) were not efficient enough to give reliable results at the extreme amplification level employed. Efficiencies were neither tested nor optimized. Further doubts about the accuracy of the quantifications stem from the purity of the templates. Removing the discussion of RNA purity from the paper does not alleviate the potential for organic contaminants from extraction to compromise cDNA synthesis efficiency. This should be tested. Finally, since the integrity of the dsRNA was not assessed, it is unclear how long the different treatments actually resulted in exposure of the roots to the dsRNAs. In the absence of measures to demonstrate the methods employed could yield quantitative and representative data, the reliability of the results remain questionable to me.